# Characterizing Disabilities in Preschool Aged Children with Congenital Zika Virus Syndrome with the ICF Model

**DOI:** 10.3390/v14102108

**Published:** 2022-09-23

**Authors:** Laura Cristina Machado Ribeiro de Souza, Carla Trevisan Martins Ribeiro, Tatiana Hamanaka, Luciana Castaneda Ribeiro, Nathália Crsitina Oliveira de Souza, Sheila Moura Pone, Karin Nielsen-Saines, Elizabeth B. Brickley, Maria Elisabeth Lopes Moreira, Marcos Vinicius da Silva Pone

**Affiliations:** 1National Institute of Women, Children and Adolescents Health Fernandes Figueira, Oswaldo Cruz, Foundation (IFF-Fiocruz), Avenida Rui Barbosa, 716, Flamengo, Rio de Janeiro 22250-020, Brazil; 2Federal Institute of Rio de Janeiro (IFRJ), Rua Professor Carlos Wenceslau, 343-Realengo, Rio de Janeiro 21710-240, Brazil; 3David Geffen School of Medicine, University of California, 10833 Le Conte Ave, Los Angeles, CA 90095, USA; 4London School of Hygiene & Tropical Medicine, Keppel St., London WC1E 7HT, UK

**Keywords:** Zika virus Infection, Congenital Zika Syndrome, microcephaly, functioning, International Classification of Functioning, Disability and Health (ICF)

## Abstract

Understanding functioning and disabilities in children with Congenital Zika Syndrome (CZS) is essential for health planning. We describe disabilities present in children with CZS followed in a reference hospital in Rio de Janeiro, Brazil, based on the biopsychosocial model of the International Classification of Functioning, Disability and Health (ICF). This was a cohort study of children >3 years of age with CZS. Disability was characterized through outcomes related to ICF components assessed via clinical and motor development evaluations. Among 50 children, with a median age of 40 months, 47 (94%) presented with severe impairment and 46 (92%) had microcephaly. Damage to the head and neck was found in most children, with abnormal central nervous system imaging universally present. Most children had cognitive impairment (92%), muscle tone problems (90%), and speech deficits (94%). We found movement limitations in all categories but more pervasively (80–94%), in postural transfers and displacements. The main environmental factors identified in the ICF model were the use of products or substances for personal consumption and access to health services. Children with CZS have extremely high rates of disability beyond aged 3 years, particularly regarding motor activity. ICF-based models can contribute to the assessment of health domains.

## 1. Introduction

Congenital Zika Syndrome (CZS) is characterized by clinical and radiological changes to the central nervous, musculoskeletal, and ophthalmologic systems, raising concerns about long-term repercussions of these findings in the lives of affected children, their families, and society [1]. The characteristic neurotropism of Zika virus (ZIKV) impairs brain development and disrupts synaptogenesis, interfering in the migration and multiplication of central nervous system cells, leading to a wide spectrum of manifestations which go beyond structural malformations. Cognitive, sensory, visual, and communication impairments are often present, directly affecting the performance of activities and social participation [2,3,4,5,6].

Due to their complex clinical needs, children with CZS and their families require coordination of health services, education, and social assistance to promote improvement of the child’s functioning across a range of life activities [7]. However, few studies to date have described long-term clinical repercussions of CZS in preschool aged children, especially their level of functioning and degree of disability [3,7].

The World Health Organization (WHO) conceptualizes functioning and disability as dynamic interactions between health conditions and contextual (i.e., personal and environmental) factors. Whereas functioning can be considered a more neutral term describing activities “either by the body or a part of the body, the whole person, or the whole person in a social context,” disability has a negative connotation and represents “impairments, activity limitations and restriction in participation” [8]. To operationalize this conceptual model (Figure 1), the WHO developed the International Classification of Functioning, Disability and Health (ICF) [8]. The ICF reflects a global shift away from exclusively focusing on morbidity towards a broader and more multidimensional view of health. The ICF provides a standardized tool for facilitating communication between the multidisciplinary professional team, which is inherent to the care of chronically sick children, such as those with CZS [8,9].

Since CZS is a relatively new disease, prior research studies have barely used the ICF model to describe the characteristics of children with CZS [3,10]. Studies published to date used the ICF model through the PC core set to assess children with SCZ [3], presenting parental perspectives on relevant areas of functionality and disability that should be included as outcome measures [10] or to describe challenges related to the implementation of the ICF in rehabilitation services for children with CZS [11]. However, knowledge on the prevalence and level of disability in children entering the preschool phase is essential for the planning of public health policies [8,11]. To address this gap, the present study aims to describe disability, based on the ICF biopsychosocial model, in children with CZS who were followed in a reference hospital in the state of Rio de Janeiro, Brazil.

## 2. Materials and Methods

This is a retrospective cohort study conducted in a dedicated multi-disciplinary outpatient clinic for follow-up of children born to pregnant women exposed to ZIKV at a reference hospital in the state of Rio de Janeiro. Children were clinically evaluated in the outpatient pediatric infectious diseases clinic by a team of specialists. Standardized instruments, such as the Gross Motor Function Measure (GMFM) and the Gross Motor Function Classification System (GMFCS), were used for evaluation of motor development in the patient population.

Study participants were children over 3 years of age with CZS who participated in the “Vertical exposure to Zika virus and its consequences for child neurodevelopment” cohort study (NCT03255369). Participants had a clinical phenotype compatible with congenital ZIKV infection following clinical-radiological evaluations, according to criteria described by Moore and collaborators (2017), regardless of laboratory confirmation. When possible, laboratory confirmation of maternal ZIKV infection during pregnancy via reverse transcription polymerase chain reaction (RT-PCR) in maternal blood, urine, placenta or amniotic fluid samples; and/or RT-PCR positivity or a positive immunoglobulin M (IgM) result in infant specimens (blood, urine or cerebrospinal fluid) was obtained. Clinical features of CZS were necessary for children to be included in the present analysis. All children were followed longitudinally at a multi-disciplinary outpatient clinic at a referral hospital in the state of Rio de Janeiro. Children with incomplete medical records (i.e., without complete information available in medical records and standardized clinical assessments) were excluded.

Ethical approval for the parent cohort study was provided by the institutional review board of IFF-Fiocruz (Plataforma Brasil, CAAE: 52675616.0.0000.5269). Parents or guardians provided written informed consent for their children’s participation.

Disability was investigated based on the ICF characteristics of body structure impairment and body function impairment, activity limitation, and restriction in participation. Contextual and environmental factors were also described. A data collection form was developed based on the findings of two prior studies: (i) an analysis identifying the common content between ICF and medical evaluation instruments in the outpatient clinic at the chosen reference hospital [12] and (ii) another analysis identifying the common content between ICF and GMFM [13].

Considering that the link between GMFM and ICF produced duplicate codes, forms were reviewed by two physical therapists familiar with both types of evaluations to ensure only one entry into the ICF categories was performed. Selection criteria included: (a) use of items evaluated in the GMFM test which serve as a cut-off for functional classification according to the GMFCS (GMFCS Diamonds) [14]; (b) selection of categories with items most representative of functioning (what would be most functional for children within the perspective of their development); (c) highest qualified bimanual or in two directed activities; and (d) transformation of some categories into fourth-level ICF categories.

It should be highlighted that GMFM is a scale for evaluating the gross motor performance of children with cerebral palsy from five dimensions, namely: A—Lying and Rolling, B—Sitting, C—Crawling and Kneeling, D—Standing, and E—Walking, Running and Jumping. The items in each dimension are scored on a four-point scale ranging from zero to three, in which: “zero” = does not initiate the required task; “one” = initiates but completes less than 10% of the required task or as indicated in the criterion descriptions; “two” = partially completes (10% to <100%) the task or does so as indicated in the criterion descriptions; “three” = completes the task as indicated [14]. This study used GMFM-88, which is the complete evaluation, and its classification [14].

For categories linked to GMFM, disability extent was also described by ICF qualifiers, as shown in Table 1. This was only possible for activities and participation, according to the GMFM grading of evaluable items (Table 1).

Secondary data describing the presence of impairments, limitations, restrictions and/or barrier experienced by study participants with CZS were obtained from clinical medical records and physical therapy evaluations. Demographic and clinical data such as sex, age, functional level (via GMFCS), and presence of microcephaly were collected from medical records and descriptively recorded in the database. GMFCS was divided into three levels: severe (GMFCS V and IV), moderate (GMFCS III), and mild (GMFCS I and II).

Epi Info™ version 7.2.3.1 was used for statistical analyses. Data were described as absolute frequencies for categorical variables and as means and standard deviations or medians and interquartile ranges (IQRs) for continuous variables.

## 3. Results

Among the 50 participating children (median age, IQR: 40 months, 36–50 months), 27 (54%) were female and 46 (92%) had microcephaly noted in their first head circumference measurement. In total, 47 (94%) children were classified at the severe functional level (GMFCS V and IV); 1 (2%) at the moderate level (GMFCS III); and 2 (4%) at the mild level (GMFCS I and II).

Structural brain impairment identified by brain imaging was universally present in the study sample (n = 50, 100%). Among children with available information, abnormalities of the head and neck were present in 100% (n = 49/49) of children on physical exam and included microcephaly, palpable sutures, occipital prominence, and craniofacial disproportion (Table 2). Abnormalities of the skin, hips and eyes were present in 81% (n = 38/47), 65% (n = 31/48) and 60% (n = 29/48) of participants, respectively. Abdominal wall (gastric or intestinal) abnormalities (e.g., umbilical/inguinal hernias) were present in 4% (n = 2/48) of participants (Table 2).

Body function evaluations demonstrated that speech fluency and rhythm (n = 47/50; 94%) and cognitive function (i.e., the state of awareness and alertness, including the clarity and continuity of the wakeful state) (n = 46/50; 92%) were frequently impaired. Among children with available information, most (n = 45/47; 96%) had impaired muscle tone (predominantly hypertonic) and impaired urinary function (i.e., including neurogenic or overactive bladders) (n = 40/43; 93%). In addition, 54% (n = 26/48) had impaired weight maintenance (i.e., difficulty gaining weight). Data were lacking on metabolic functions (n = 27, 54% missing), motor reflexes (n = 26, 52% missing), and joint mobility (n = 33, 66% missing) (Table 2).

In the analysis of activity and participation, almost all children had complete limitations, characterized by an inability to initiate the proposed movement in maintenance activities or in achieving motor milestones, such as kneeling (n = 47/50; 94%), maintaining body position (n = 46/50; 92%), and standing (n = 47/50; 94%). Most children were also unable to perform activities related to displacement such as crawling and walking, and postural transfers (90% and 98%, respectively), except rolling. A good part of the sample (n = 20; 40%) was able to initiate the activity or complete less than half of the movement, being classified as severe and not complete (Table 3). In fact, most children had mildly to severely limited performance of simple activities, such as using their hands and arms and reaching out. The majority were only able to perform activities related to head control such as shifting the body’s center of gravity and maintaining head position. Data on participation was not available in the GMFM or in clinical assessment forms (Table 3).

Regarding environmental factors, medical records lacked information as to the use of assistive mobility devices (wheelchairs and walking devices) and use of orthotic devises for 49 (98%) and 48 (96%) children, respectively (Table 4). In regard to other products or substances used by study participants, 98% (n = 49) of children used medications such as anticonvulsants or drugs for overactive bladders. Access to health services, systems, and policies was available for most children. The records indicated that 78% of children (n = 39/50) underwent physical therapy, 66% (n = 33/50) had access to speech therapy, and 94% (n = 45/48) had consultations with a pediatric neurology service (Table 4).

## 4. Discussion

Children prenatally exposed to ZIKV, especially those with CZS, present with multiple disabilities that persist over time and directly impact their care. Current evidence suggests that the disabilities of children with CZS who are over three years of age do not improve with time and reflect important limitations in activity and are influenced by environmental factors. However, these limitations are not well described in the literature since most studies emphasize structural defects and body part-specific functional impairments [6,15].

There was no evidence of significant sex-related differences in our study population, which is consistent with prior studies [5,7,16]. The median age of our study population was 40 months (three years and four months). Thus, these children were in the preschool phase and at an older age than most children evaluated in prior studies [3,5,10,17]. Recent studies, such as the one conducted by Campos and collaborators [10], addressed functionality in children with CZS whose mean age was 32.2 months; another study conducted by Cavalcante and collaborators evaluated growth and motor development in children with CZS who were up to 36 months of age [5]. An important limitation of the current analysis was the markedly high frequency of microcephaly within the cohort, which limits generalizability of the current findings to children experiencing more mild presentations of CZS.

A high number of children were considered to be in the severe category according to the GMFCS; this is consistent with other studies evaluating the severity of motor function deficits in children with CZS [16,18,19]. The high number of children with severe CZS (GMFCS V and IV) in the present cohort reinforces the notion that these patients maintain functional impairment as they age. Thus, important disabilities are to be expected not only regarding structural and functional damages but also serious or complete limitations in activity and participation. Our findings underscore the pressing need for comprehensive assistance and care planning for these children and their families [20].

Regarding structural disabilities, the literature demonstrates that microcephaly and redundant scalp skin are typical findings in children with CZS, as are cerebral and ophthalmologic changes [2,5,15]. Structural brain abnormalities are well described in the CZS literature [15,21]. Cavalcante and collaborators reported that all children in their study population had brain imaging abnormalities, of which intracranial calcifications (93.5%), reduced cerebral parenchyma (85.8%), and malformations of cortical development (78.3%) were the most prevalent [5]. Moreover, eye lesions such as chorio-retinal atrophy, retinal pigment spots, macular scarring, and glaucoma are well documented ocular abnormalities which help define CZS [1,2,22,23]. Changes in upper and lower limbs are also described in the literature, particularly acquired hip dislocation [5,24]. Abdominal abnormalities tend to be rare, as was noted in the present cohort [4].

The literature also documents changes to fluency and speech rhythm, consciousness, muscle tone, and urinary function as the major bodily functional disabilities in CZS [3,25]. Studies in children with CZS describe muscle hypertonia and consciousness-altering epilepsy [5,6,16,26]. Nevertheless, fluency and rhythm of speech are poorly discussed in the literature, despite being highly relevant for the future social inclusion and participation of individuals with CZS. Ferreira and collaborators found important changes in mental language functions [3] and Carvalho and collaborators observed language impairment in 95.7% among 69 children with CZS [16]. Monteiro and collaborators found damage to the urinary function in all of their patients with CZS [25].

It is important to highlight that respiratory changes were present in a quarter of our patients. The literature reports that children with CZS may develop symptoms similar to those found in other children with cerebral palsy, such as ineffective cough and respiratory disorders due to muscle tone and strength changes [27]. The evaluation of this domain is important since respiratory impairments can increase the degree of morbidity, even causing the need for hospitalization and ventilatory support [27] and/or mortality [28].

A key limitation of the study was the reliance on existing medical and physical therapy records for disability-related data. Due to limitations in the data recorded, we were not able to evaluate the “visual function” item based on medical records nor participation based on physical therapy records, which focused on GMFM evaluations that did not include these assessments. Nevertheless, GMFM was able to identify and evaluate several limitations in activity, especially those related to locomotion and postural transfers.

A study conducted in Northeastern Brazil, which evaluated 24-month-old children with CZS via GMFM, reported results similar to ours, despite the sample being younger [4]. Most of our children (96%) were in the severe category (with enough impairments in simple motor functions of GMFM dimensions A (lying and rolling) and B (sitting). Only children in the light group achieved some unspecified displacement and postural transfers [4]. In a study by Takahasi and collaborators, conducted in the northeastern state of Brazil, most children with severe CZS showed no improvement in gross motor function in their third year of life; the authors concluded that they probably had already approached their maximum motor function potential [29]. This is consistent with our present findings. 

Results on ICF qualifiers indicate that most disabilities related to activity are qualified as complete limitations. In the literature, only one study by Ferreira and collaborators used ICF activity and participation qualifiers. In their sample of 34 children with CZS (mean age 21.2 months), all children had limited body movement, with severe impairment noted in 65%. None of the children attained gait, a complete limitation noted in almost 70% of evaluable children, being not evaluable in the remainder due to their younger age [3]. 

Regarding contextual (i.e., environmental and personal) factors, we found gaps in documentation across several categories, likely due to the multidisciplinary structure of the outpatient clinic, where different specialties use medical records tailored to their focus area. In this sense, we could not identify all contextual factors, especially use of assistive devices, as we lacked information on family relationships, support networks and personal factors, which is also a study limitation. Further research regarding social networks is warranted and will be very important since it will enable healthcare providers to assess issues which go beyond the noted disabilities, thus facilitating support in a way which can improve care and day-to-day functioning of these children [30]. 

The use of products and substances for personal consumption (e.g., medications) and health systems can be easily explained by these children having access to medications and multi-disciplinary follow-up since they receive care in a reference hospital in Rio de Janeiro. The Study by Ferreira et al. [3] showed that parents of children with CZS identify access to medication and therapy as facilitating environmental factors. Prior research (10) has evaluated the perception of parents regarding the needs of their children as guided by the ICF model. Although parents emphasized the relevance of mobility issues, the availability of environmental factors (especially access to health services and systems) enabled a healthier family lifestyle and improved their children’s quality of life as perceived by parents and guardians.

The use of the ICF model in CZS still faces serious challenges. The use of the Cerebral Palsy Core Set (as in prior research) [3], seems to greatly restrict evaluation of multiple features present in CZS. Furthermore, according to Longo and collaborators [11], the biggest challenges in the field relate to the lack of standardized measurement instruments, low training of health teams in evaluating activity, participation, and contextual factors in rehabilitation, and the lack of an ICF-based record system within the Brazilian unified health system. Implementation of an ICF model within this context will require further research and investment. However, it is clear from the present study that the child with CZS should be more broadly assessed using multiple components as proposed in the ICF model and not just evaluated for deficiencies in structure and function.

## 5. Conclusions

This study showed that disabilities of children with CZS go far beyond body structure and body function. The severe damage documented in initial studies, especially to the central nervous and ophthalmologic systems, persist over time, regardless of the use of rehabilitation services by most patients. The limitations found in activity (especially mobility) directly imply very restricted social participation from these children, a feature not easily evaluable from medical records. Environmental factors function as facilitators or barriers to activity and participation.

Very few studies to date have correlated CZS and ICF; the few studies that used the GMFM scale failed to describe qualifiers of this scale in their results, which was performed in the present analysis. Our study supports recent reports [14,15] highlighting the existing challenges of implementing the ICF model in CZS [3,10,11]. Thus, we conclude that the ICF-based biopsychosocial model of universal care can contribute to the assessment of all domains, helping address the practical challenges experienced by professionals and family members of children with CZS.

## Figures and Tables

**Figure 1 viruses-14-02108-f001:**
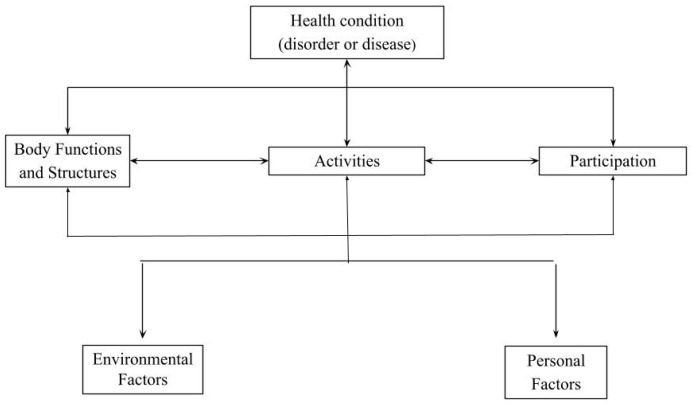
ICF Model: Interactions between the components Adapted [8].

**Table 1 viruses-14-02108-t001:** Correlation between GMFM Qualifiers and ICF.

GMFM Qualifier	ICF Qualifier *
0 = does not perform the movement	4 = complete limitation
1 = initiates the movement	3 = severe limitation
2 = partially completes the movement	1 = mild limitation
3 = completes the movement	0 = no limitation

Note = * Qualifier 2 (moderate limitation) will not be used; GMFM = Gross Motor Function Measure; ICF = International Classification of Functioning, Disability and Health.

**Table 2 viruses-14-02108-t002:** Characterization of body structure and functional impairments in preschool aged children with CZS (n = 50).

Category (ICF Code)	Abnormality		No Information
	Yes	No	
Body structure			
Trunk (s760)	8 (16%)	36 (72%)	6 (12%)
Head and neck region (s710)	49 (98%)	-	1 (2%)
Skin areas (s810)	38 (76%)	9 (18%)	3 (6%)
Eye socket (s210)	8 (16%)	36 (72%)	6 (12%)
Mouth (s320)	8 (16%)	36 (72%)	6 (12%)
Immune system—lymph nodes (S420)	-	10 (20%)	40 (80%)
Stomach (s530)	-	47 (94%)	3 (6%)
Intestine (s540)	2 (4%)	46 (92%)	2 (4%)
Pancreas (s550)	-	48 (96%)	2 (4%)
Liver (s560)	-	48 (96%)	2(4%)
Gall bladder and ducts (s570)	-	48 (96%)	2 (4%)
Spleen (s4203)	-	48 (96%)	2 (4%)
Hip joint (s75001)	31 (62%)	17 (34%)	2 (4%)
Reproductive system—genitalia (s630)	12 (24%)	33 (66%)	5 (10%)
Upper extremity (S730)	12 (24%)	37 (74%)	1 (2%)
Lower extremity (s750)	17 (34%)	32 (64%)	1 (2%)
Vertebral column (s7600)	8 (16%)	33 (66%)	9 (18%)
Eyeballs—FO (s220)	29 (58%)	19 (38%)	2 (4%)
Brain (s110)	50 (100%)	-	-
**Body functions**			
General metabolic (b540)	7 (14%)	16 (32%)	27 (54%)
Hearing (b230)	8 (16%)	39 (78%)	3 (6%)
Cognitive function (b110)	46 (92%)	4 (8%)	-
Swallowing (b510)	20 (40%)	6 (12%)	24 (48%)
Muscle tone (b735)	45 (90%)	2 (4%)	3 (6%)
Motor reflex (b750)	13 (26%)	11 (22%)	26 (52%)
Joint mobility (b710)	6 (12%)	11 (22%)	33 (66%)
Fluency and rhythm of speech (b330)	47 (94%)	3 (6%)	-
Weight maintenance (b530)	26 (52%)	22 (44%)	2 (4%)
Urinary (b620)	40 (80%)	3 (6%)	7 (14%)
Respiratory (b440)	13 (26%)	37 (74%)	-
Heart (b410)	1 (2%)	49 (98%)	-
Heart (b410) Echocardiogram	4 (8%)	45 (90%)	1 (2%)

Note: ICF = International Classification of Functioning, Disability and Health; FO = eye fundus.

**Table 3 viruses-14-02108-t003:** Characterization of limitations observed in preschool aged children with CZS (n = 50).

Category (ICF Code)	Limitation				
	No	Yes			
	Qualifier				Total Limitations *
	0	1	3	4	
Hand and arm use (d445)	14 (28%)	19 (38%)	17 (34%)	-	36 (72%)
Reaching out (d4452)	13 (26%)	11 (22%)	24 (48%)	2 (4%)	37 (74%)
Rolling over (d4107)	10 (20%)	1 (2%)	20 (40%)	19 (38%)	40 (80%)
Shifting the body’s center of gravity (d4106)	28 (56%)	12 (24%)	6 (12%)	4 (8%)	22 (44%)
Changing basic body position (d410)	9 (18%)	2 (4%)	12 (24%)	27 (54%)	41 (82%)
Sitting (d4103)	6 (12%)	-	6 (12%)	38 (76%)	44 (88%)
Maintaining head position (d4155)	30 (60%)	9 (18%)	5 (10%)	6 (12%)	20 (40%)
Maintaining a sitting position (d4153)	8 (16%)	1 (2%)	8 (16%)	33 (66%)	42 (84%)
Lying down (d4100)	5 (10%)	-	2 (4%)	43 (86%)	45 (90%)
Transferring oneself while sitting (d4200)	3 (6%)	2 (4%)	-	45 (90%)	47 (94%)
Moving around (d455)	5 (10%)	1 (2%)	2 (4%)	42 (84%)	45 (90%)
Maintaining body position/4 supports (d415)	4 (8%)	1 (2%)	1 (2%)	44 (88%)	46 (92%)
Crawling (d4550)	3 (6%)	-	-	47 (94%)	47 (94%)
Kneeling (d4102)	3 (6%)	1 (2%)	1 (2%)	45 (90%)	47 (94%)
Standing (d4104)	3 (6%)	-	-	47 (94%)	47 (94%)
Maintaining a standing position (d4154)	3 (6%)	-	-	47 (94%)	47 (94%)
Squatting (d4101)	3 (6%)	-	-	47 (94%)	47 (94%)
Bending (d4105)	3 (6%)	-	-	47 (94%)	47 (94%)
Walking (d450)	3 (6%)	-	-	47 (94%)	47 (94%)
Carrying objects in the hands (d4301)	3 (6%)	-	-	47 (94%)	47 (94%)
Walking around obstacles (d4503)	1 (2%)	-	1 (2%)	48 (96%)	49 (98%)
Running (d4552)	1 (2%)	-	1 (2%)	48 (96%)	49 (98%)
Kicking (d4351)	1 (2%)	-	-	49 (98%)	49 (98%)
Jumping (d4553)	1 (2%)	-	-	49 (98%)	49 (98%)
Climbing (d4551)	1 (2%)	-	-	49 (98%)	49 (98%)

Note: 0 (without limitation), 1 (mild limitation), 3 (severe limitation), and 4 (complete limitation); * Sum of the percentages of qualifiers 1, 3 and 4; ICF = International Classification of Functioning, Disability and Health.

**Table 4 viruses-14-02108-t004:** Sample distribution according to environmental factors (n = 50).

Categories (ICF Encoding)	Makes Use of		No Information
	Yes	No	
Products and technology for personal indoor and outdoor mobility and transportation (e120)	-	1 (2%)	49 (98%)
Products and technology for personal use in daily living—orthosis (e115)	2 (4%)	-	48 (96%)
General social support services, systems and policies—government assistance (e575)	10 (20%)	36 (72%)	4 (8%)
Products and technology for personal use in daily living—GTT (e115)	15 (30%)	33 (66%)	2 (4%)
Assistive products and technology for communication—glasses (e1251)	27 (54%)	20 (40%)	3 (6%)
Products or substances for personal consumption—Medication (e110)	49 (98%)	1 (2%)	-
Health services, systems and policies—Physical therapy (e580)	39 (78%)	11 (22%)	-
Health services, systems and policies—Speech Therapy (e580)	33 (66%)	17 (34%)	-
Health services, systems and policies—Neurology (e580)	45 (90%)	3 (6%)	2 (4%)

Note: ICF = International Classification of Functioning, Disability and Health; GTT = gastrostomy.

## Data Availability

Data for this study is available upon request.

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
