# Peer review of "Characterizing Disabilities in Preschool Aged Children with Congenital Zika Virus Syndrome with the ICF Model"

_viruses, 2022, doi:10.3390/v14102108_

Round 1

Reviewer 1 Report

Thank you for the opportunity to review this manuscript. I am glad that the authors are continuing to follow their cohort closely with CZS so we can understand preschool age outcomes. Some areas that I felt could be addressed to improve the manuscript include:  

1  Abstract- Some incomplete sentences make it hard to read. The term “environmental factors” in line 26 of abstract seems out of place since it in not presented earlier in the abstract.

2   Introduction may be good to include reference to other studies of CZS that have also used the ICF model.  These are mentioned in the Discussion, but I think might also do well to introduce earlier in the manuscript.

3.       The ICF model and more specifics of what it includes should be added to the Introduction.

4.       Methods should include exclusion criteria.

5.       Results section- Were there any children ineligible? It states, “50 children participating”, but this was a retrospective cohort study. Were they participating in the clinical follow-up program?  This just seems a little confusing as to these specific 50 children.

6.       Results section- 92% had microcephaly- was this all prenatal onset microcephaly, or some postnatal?  Was this based on HC measurement during the followup evaluation?

7.       Does the ICF model include feeding? I did not see that in the body structure of body functions listed in table 2, but certainly is an important function and many children with CZS have impaired oral feeding skills.

8.       It is unfortunate that most children do not have information on assistive devices.  This should be added to the study limitations in the Discussion section.

9.       Discussion- first line “especially those with CZS, present with several disabilities…” I think this could instead say, “especially those with CZS, present with multiple disabilities…”

1   Discussion- second paragraph states, “There was no evidence of significant sex-related differences…”  This was not presented in the results section so should not be first discussed in the Discussion section.

1   Can the authors discuss how use of the ICF model can improve care?  Can they make a recommendation for the type of data regarding body structure and function that should be collected at all follow-up appointments to standardize care and ability to understand outcomes? A better description of the added benefit o using the ICF model to the other descriptions of disability and GMFCS, etc. would be helpful for readers to better understand the significance.

Author Response

Dear Reviewer,Thank you for carefully reviewing our manuscript. Our text was reviewed and modified as requested.In response to the comments made by reviewer 1, we would like to clarify each point a bit further.

  1. Answer: Environmental factors are an assessment component of the ICF model as are body structures and functions and activity. The word limit prevented us from describing the model in greater depth, but we revised the abstract and further detailed the introduction to better explain the ICF model in order to improve reader comprehension.
  2. Answer: Lines 63 and 64 of the introduction reference other studies that are using the ICF model in the assessment of children with congenital ZIKA syndrome, however, we further detailed these studies in the discussion. In response to the reviewer’s request, we have added some information about these articles in the introduction section (lines 64-67).
  3. Answer: To better explain the model we have added Figure 1 which details  the ICF model in an illustrative way. We also have expanded the description of the model in the introduction section.
  4. Answer: The exclusion criteria is now described in lines 90-93.
  5. Answer: All children in our cohort, who were 3 years of age or more were evaluated, unless they had incomplete medical records and absence of assessments as explained in the inclusion criteria. The cohort included all children receiving clinical follow-up at our institution.
  6. Answer: Yes. 92% of children had microcephaly in the first HC measurement obtained, so these were cases of primary microcephaly. We have clarified this in line 142.
  7. Answer: Eating is an activity, however it was not described as it is not an activity contained in the GMFM tool. Swallowing is a function, and this was described in table 2.
  8. Answer: We agree with the reviewer and have included this information as a study limitation in lines 274-276. The text was modified to clarify this point.
  9. Answer: We have changed the text as suggested by the reviewer.
  10. Answer: In the results section we presented the frequency of gender (line 142) 54% of participants were female. The population was evenly divided as to gender. We thought it would be important to highlight that the literature does not demonstrate a specific gender predominance in children with CZS.
  11. Answer: There is a brief discussion of the benefits of the ICF model in the last paragraph of the discussion. There is also a reference to the ICF model in the second paragraph of the conclusion. We had not expanded on this topic because the research objective was not consistent with the use of the ICF model in clinical practice. However, we definitely agree that it is very important for children with CZS to be evaluated within the context of the ICF model. We added additional text to the end of the discussion (lines 297-299).

Reviewer 2 Report

Descriptive study on disabilities in preschool aged children with congenital zika virus syndrome using a differentiated approach, the ICF model. The study brings very relevant contributions and the limitations are presented. I suggest improving the description of sample and the origin cohort.

Author Response

Dear Reviewer,

Thank you for carefully reviewing our manuscript.

Our text was revised and modified as requested. We have included the additional information requested by the reviewer (lines 90-93), while expanding and clarifying the text.